# A Low Number of Baselines γδ T Cells Increases the Risk of SARS-CoV-2 Post-Vaccination Infection

**DOI:** 10.3390/vaccines12050553

**Published:** 2024-05-18

**Authors:** Juan Carlos Andreu-Ballester, Lorena Galindo-Regal, Carmen Cuéllar, Francisca López-Chuliá, Carlos García-Ballesteros, Leonor Fernández-Murga, Antonio Llombart-Cussac, María Victoria Domínguez-Márquez

**Affiliations:** 1FISABIO Foundation, 46020 Valencia, Spain; loregal@msn.com (L.G.-R.); lopez_frachu@gva.es (F.L.-C.); 2Parasitic Immunobiology and Immunomodulation Research Group (INMUNOPAR), Complutense University of Madrid, 28040 Madrid, Spain; cuellarh@ucm.es; 3Laboratory of Molecular Biology and Research Department, Arnau de Vilanova University Hospital, FISABIO Foundation, 46015 Valencia, Spain; garcia_carbar@gva.es; 4Microbiology and Parasitology Department, Complutense University, 28040 Madrid, Spain; 5Hematology Department, Arnau de Vilanova Hospital, 46015 Valencia, Spain; 6Medicine Department, Cardenal Herrera University, 46115 Valencia, Spain; 7Molecular Oncology Laboratory, Arnau de Vilanova Hospital, 46015 Valencia, Spain; malefermu67@gmail.com; 8Oncology Department, Arnau de Vilanova Hospital, 46015 Valencia, Spain; allombart1@yahoo.com; 9Microbiology Department, Arnau de Vilanova Hospital, 46015 Valencia, Spain; dominguez_vicmar@gva.es

**Keywords:** SARS-CoV-2, vaccine, antibodies, αβ T cells, γδ T cells

## Abstract

**Background:** The COVID-19 pandemic is the biggest global health problem in the last hundred years. The efficacy of the vaccine to protect against severe disease is estimated to be 70–95% according to the studies carried out, although there are aspects of the immune response to the vaccine that remain unclear. **Methods:** Humoral and cellular immunity after the administration of three doses of the Pfizer–BioNTech and Oxford AstraZeneca vaccines against SARS-CoV-2 over one year and the appearance of post-vaccination COVID-19 were studied. SARS-CoV-2 IgG and IgA antibodies, αβ and γδ T-cell subsets, and their differentiation stages and apoptosis were analyzed. **Results:** Anti-SARS-CoV-2 IgG and IgA antibodies showed a progressive increase throughout the duration of the study. This increase was the greatest after the third dose. The highest levels were observed in subjects who had anti-SARS-CoV-2 antibodies prior to vaccination. There was an increase in CD4+ αβ, CD8+ γδ and TEM CD8+ γδ T cells, and a decrease in apoptosis in CD4+ CD8+ and CD56+ αβ and γδ T cells. Post-vaccination SARS-CoV-2 infection was greater than 60%. The symptoms of COVID-19 were very mild and were related to a γδ T cell deficit, specifically CD8+ TEMRA and CD56+ γδ TEM, as well as lower pre-vaccine apoptosis levels. **Conclusions:** The results unveil the important role of γδ T cells in SARS-CoV-2-vaccine-mediated protection from the disease.

## 1. Introduction

The COVID-19 pandemic is the first major global health issue that has emerging in a little over a century in a way not seen since the 1918 H1N1 pandemic. This caused a great effort in biomedical research to obtain a vaccine that could stop or improve the evolution of the infection worldwide [1].

There was high vaccine hesitancy during the COVID-19 pandemic (close to 30%) due to, among other reasons, the doubts about the efficacy of the vaccine [2]. The effectiveness of full vaccination against SARS-CoV-2 infection was estimated to be from 70.4% by the ChAdOx1 nCoV-19 vaccine (AZD1222; Oxford-AstraZeneca) to 95% by the BNT162b2 mRNA COVID-19 vaccine (Pfizer-BioNTech). When the prevention of hospitalization and intensive care unit admission and severe disease were assessed, the effectiveness of vaccination ranged between 89 and 93% [3,4,5]. The WHO suggested that a clear demonstration of efficacy should be a minimum criterion for any acceptable COVID-19 vaccine. The efficacy can be assessed against disease, severe disease, and/or shedding/transmission, and the standardization of quantifiable endpoints is still a challenge [6]. 

Several studies identified a significant correlation between the dynamics of antibodies and the effectiveness of COVID-19 vaccination. Likewise, there is a correlation between anti-SARS-CoV-2 antibodies and protection from infection, as well as antibody levels and the likelihood of transmission. In a general manner, the peak humoral responses were reached at 3–4 weeks post second dose of messenger RNA (mRNA) vaccines such as Pfizer-BioNTech (mRNA BNT161b2) and Moderna (mRNA-1273), after which the antibody levels progressively decreased at 120–180 days post vaccination. The significant decline in anti-SARS-CoV-2 antibodies over time shows the progressive decline in the vaccine efficacy for preventing SARS-CoV-2 infections, although with a reduction in the risk of developing severe COVID-19. This decay of specific antibodies supports the need for boosters [7,8,9]. Protective and efficient humoral immune responses, with SARS-CoV-2-neutralizing antibodies, developed after the second or third dose of COVID-19 vaccination, with a later decrease [10]. Although a significant decrease in humoral and cellular responses after 6 months of vaccination was recently reported [11], there are still many questions to be resolved about COVID-19 vaccines, such as the duration of humoral immunity after primary infection or long-term vaccination. Similarly, knowledge about the duration of protection conferred by booster injections is also limited [12].

Cellular immunity plays an important role in limiting disease severity and the resolution of SARS-CoV-2 virus infection. T cells have an important role in protecting against SARS-CoV-2 infection, and vaccination leads to the development of both humoral and cellular immunity against the spike protein. The presence of SARS-CoV-2-specific CD4+ and CD8+ T cells was associated with a reduced disease severity. Moreover, the immunogenicity of SARS-CoV-2 vaccines involves a cellular response, but the phenotypic and functional diversity of the T-cell subsets involved in vaccine protection and infection control is not well understood [13]. Inactivated vaccines with peptides from SARS-CoV-2 proteins elicited specific humoral and cellular responses. The frequencies of CD4+ T cells producing IFN-γ, IL-2, and TNF-α in response to SARS-CoV-2 spike, nucleocapsid, or membrane proteins were significantly higher in double-vaccinated subjects [14].

T cell differentiation in memory T-cell subsets plays a major role in the effectiveness of vaccines. For these reasons, in this work, naïve T cells (TN), central memory T cells (TCM), peripheral effector memory T cells (TEM), and terminal effector memory T cells (TEMRA) were studied [15].

The main objective of this work was the study of humoral and cellular immunity after the administration of the Pfizer–BioNTech and Oxford AstraZeneca vaccines against SARS-CoV-2. For this, specific IgG and IgA antibodies and αβ and γδ T cells were studied. The study was conducted over a period of one year and after three doses of vaccines. Subsequently, the relationship of the results obtained with the appearance of post-vaccination COVID-19 was studied.

## 2. Materials and Methods

### 2.1. Ethics Statement 

The Research Ethics Committee of Arnau de Vilanova hospital, Valencia (Spain) approved the study (10/2021-March 24). Each volunteer participant signed an informed consent document. This study was conducted following the recommendations of the Spanish Bioethics Committee, the Spanish legislation on Biomedical Research (Law 14/2007) and Personal Data Protection (Spanish Law 3/2018 and European Law UE676/2018). The anonymity of the subjects participating in the study has been ensured.

### 2.2. Study Population

In this prospective follow-up study, a total of 40 volunteers were recruited, followed up over time (1 year), before and after vaccination. Pfizer–BioNTech (BNT162b2) COVID-19 vaccine (mRNA) and Oxford AstraZeneca (ChAdOx1) (adenovirus modified) were used for vaccination. Serum samples were taken at five time points. SARS-CoV-2 IgG and IgA antibodies, αβ and γδ T-cell subsets and their differentiation stages were analyzed at different post-vaccination times: 0 (baseline), pre-vaccine; 1st: 1 month after the first dose; 2nd: 1 month after the second dose; 3rd: 6 months after the second dose; 4th: 1 month after the third dose (Figure 1). The entire cohort of individuals received 3 vaccine doses. Blood samples were obtained just prior to the administration of the vaccines. The study subjects had to meet a series of criteria: not suffer from or have previously suffered from COVID-19, not have had suggestive symptoms, not suffer from an infectious disease at the time of vaccination, not have a known immunodeficiency or autoimmune or neoplastic disease, not carry immunosuppressive treatment, and not having received another vaccine in the previous six months. All subjects who presented symptoms related to SARS-CoV-2 infection after vaccination were assessed using an antigen test. If the result was negative, a PCR test was performed to confirm or exclude infection.

### 2.3. Cell Isolation for Analysis of γδ and αβ T Cells 

The blood cell counts were obtained using a cell counter (LH750 Beckman Coulter, Inc., Fullerton, CA, USA).

An enrichment of the sample in mononuclear cells (MNCs) was obtained by centrifuging the EDTA anticoagulated blood sample on a density gradient using Lymphoprep™ (Palex Medical SA, Barcelona, Spain) at 3500 rpm for 20 min. After two washes in phosphate-buffered saline (PBS), the cells obtained were resuspended in 200 mL of PBS.

### 2.4. Functional Analysis of γδ and αβ T Cells

To evaluate the functional analysis of γδ and αβ T cells, the MNCs were firstly incubated in two tubes with anti-TCR PAN αβ-PE and anti-TCR PAN γδ-PE antibodies during 10 min at room temperature, and then labeled with the following monoclonal antibodies conjugated with fluorochromes (Beckman Coulter, Inc., Miami, FL, USA):

Tube 1: anti-TCR PANαβ-PE (clone: IP26A), CD19-PE (clone: J3-119), CD45RA-ECD (clone: 2H4LDH11LDB9 (2H4)), CD56-PC7 (clone N901 (NKH-1)), CD62L-APC (clone: DREG56), CD4-APC A750 (clone: 13B8.2), CD3-APC A700 (clone: UCHT1), CD8-PB (clone: B9.11), and CD45-KRO (clone: J33).

Tube 2: anti-TCR PANγδ-PE (clone: IMMU 510), CD19-PE (clone: J3-119), CD45RA-ECD (clone: 2H4LDH11LDB9 (2H4)), CD56-PC7 (clone: N901 (NKH-1)), CD62L-APC (clone: DREG56), CD4-APC A750 (clone: 13B8.2), CD3-APC A700 (clone: UCHT1), CD8-PB (clone: B9.11), and CD45-KRO (clone: J33).

After incubation for 10 more minutes at room temperature, the VersaLyse lysis agent (Beckman Coulter, Inc) was used. One mL of the “Fix-and-Lyse” mixture prepared at that time to the mononuclear cells (MNCs) was added, mixed, and incubated for 20 min at room temperature, protected from light. Then, 2 mL of PBS were added. After the samples were centrifuged for 5 min at 1500 rpm and at room temperature, the supernatants were removed by aspiration and the cell buttons were resuspended in 0.5 mL of PBS.

Acquisition and analysis were performed on a Navios flow cytometer (Beckman Coulter, Inc.) and analyzed with Kaluza Software (V. 2.2-06/07/2022). A total of 100,000 events were acquired. The absolute counts of circulating cell subsets were calculated using dual-platform counting technology.

### 2.5. Apoptosis Evaluation

Apoptosis detection was performed with an ANNEXIN V-FITC/7-AAD Kit (Beckman Coulter, Inc.), based on the binding properties of annexin V to phosphatidylserine and the specificity of 7-amino-actinomycin D (7-AAD) for the DNA guanine–cytosine base pair, following the instructions of the manufacturer. The results described in this work refer to early apoptosis (Annexin V+, 7-AAD-), which accounts for 90–95% of the total apoptosis.

### 2.6. Detection of IgG and IgA Antibodies against SARS-CoV2

The detection of IgG antibodies against SARS-CoV-2 was performed using the SARS-CoV-2 IgM and SARS-CoV-2 IgG II Quant assays, with the corresponding calibrators and controls for the ARCHITECT i2000SR analyzer (Abbott Diagnostics, Mississauga, ON, Canada). The ARCHITECT System uses chemiluminescent microparticle immunoassay (CMIA) technology as the detection method to measure and quantify the concentration of antibodies present in the serum. The SARS-CoV-2 IgG II Quant assay quantifies IgG in BAU/mL, since it is standardized according to the WHO standard. The IgG values were correlated with the WHO international standard 20/136, thus the units were expressed in BAU/mL (BAU: binding antibody units). The cut-off level of a positive IgG result was ≥7.1 BAU/mL. The levels of IgA against SARS-CoV-2 were analyzed in serum, using the DIAPRO COVID19 IgA Elisa KIT (Diagnostic Bioprobes Srl, Sesto San Giovanni, Italy) according to the manufacturer’s recommendations. The microplates were coated with immunodominant and nucleocapsid extender recombinant spike glycoproteins specific for COVID-19. The cut-off level of a positive IgA result was ≥1.1 AU/mL.

### 2.7. Statistical Analysis

A non-parametric Wilcoxon test was used to compare the evolution of antibody mean values and αβ-γδ T-cell subsets. The Mann–Whitney U test was used to compare the differences between subjects with previous natural infection (pre-infection antibodies against SARS-CoV-2 after previous vaccination (PI)) and no infection. Fisher’s exact test in contingency tables (odds ratio with CI95%) was used for qualitative variables. A *p* value < 0.05 was considered statistically significant.

The graph and statistical analyses were performed using GraphPad Prism software version 6.0 for Windows (GraphPad Software, San Diego, CA, USA).

## 3. Results

### 3.1. Subjects Studied

Forty healthy volunteers were enrolled in the present study, of which 29 were female (72.5%) and 11 male (27.5%). The mean age was 48.9 ± 10.7 years (range 26–64). Figure 1 shows the evolution of vaccination time and post-vaccination COVID-19 infection throughout the study. According to the vaccine type, 21 subjects (52.5%) received an mRNA vaccine and 19 subjects (47.5%) received an adenovirus-based vaccine.

### 3.2. IgG and IgA SARS-CoV-2-Specific Antibody Responses and CD19+ B Cells after Vaccination

When IgG and IgA antibodies against SARS-CoV-2 were analyzed prior to the first vaccine dose, five (12.5%) and nine (22.5%) patients had IgG and IgA antibodies against SARS-CoV-2, respectively (subjects with a previous natural infection). Five subjects were IgG- and IgA-positive at the same time.

The positive subjects were asymptomatic. Figure 1A shows the dynamics of IgG and IgA antibodies against SARS-CoV-2 during the course of the study (1st to 4th analysis) depending on whether or not patients were subject to a previous natural infection with SARS-CoV-2. In summary, a progressive increase in anti-SARS-CoV-2 antibodies was observed, which was greater after the third dose of the vaccine. IgG levels one month after the first dose were higher in subjects who had pre-vaccination anti-SARS-CoV-2 antibodies. Specific antibody levels tended to decrease 8 months post-vaccination, that is, 6 months from the second dose. In the case of IgG, subjects without previous natural infection showed significant increases after the first dose. Subjects who did have a previous natural infection only showed significant increases in IgG after the third dose. In the case of IgA, the vaccine only produced significant increases in subjects who did not have a previous natural infection. The vaccine did not produce any change in IgA levels in subjects who had a previous natural infection upon the administration of the first dose. In addition, these subjects showed the highest IgA levels throughout the entire follow-up. Figure 1B shows that the number of B cells during the course of the study depended on whether or not the subjects had a previous natural infection. Basal CD19+ cell levels were higher in subjects who had anti-SARS-CoV-2 antibodies prior to vaccination. In the case of IgA, this difference was statistically significant. Any change in CD19+ cell levels was not observed throughout the duration of the study.

### 3.3. Evolution of αβ and γδ T-Cell Subset Number and Apoptosis during the Course of the Study

Figure 2 shows a significant increase in the number of CD3+CD4+ αβ and CD3+CD8+ γδ T cells 6 months after administering the second dose of vaccine in subjects with a previous natural infection. In these subjects, CD3+CD4+ αβ T cells decreased to baseline levels significantly from the third dose. 

A decrease in the apoptosis of CD3+, CD3+CD4+, CD3+CD8+, and CD3+CD56+ αβ T cells was observed in subjects with a previous natural infection. These differences were highly significant at 6 months after the second dose. In the case of γδ T cells, this fact was observed with CD3+CD8+ and CD3+CD56+ subsets. In subjects without a previous natural infection, the vaccine induced an increase in apoptosis over time that was significant 6 months after the second dose. 

### 3.4. Dynamics of αβ T-Cell Differentiation Stages during the Course of the Study

A significant increase in naïve CD4+ and CD8+ αβ T-cell counts was observed throughout the course of the study in subjects without a previous natural infection. On the other hand, there was a decrease in naïve CD56+ αβ T cells after the third dose in subjects with a previous natural infection.

However, TCM CD4+ and CD8+ αβ T cells showed a decreasing trend after successive vaccine doses in subjects without a previous natural infection.

In a general manner, TEM αβ T cells were significantly decreased in subjects without a previous natural infection. In contrast, TEM CD56+ αβ T cells increased at 6- and 12-months post-vaccination in subjects without a previous natural infection.

The percentages of TEMRA CD56+ αβ T cells were significantly higher in subjects with a previous natural infection. This difference was observed before the administration of the first vaccine dose (Figure 3). 

### 3.5. Dynamics of γδ T-Cell Differentiation Stages during the Vaccination Period

The percentage of naïve γδ T-cell subsets was higher (not statistically significant) in subjects without a previous natural infection for all differentiation stages (Figure 4A). An increase in naïve double-negative and CD8+ γδ T cells was observed after the second dose. There was a decrease in the percentages of TCM CD3+ and double-negative γδ T cells 6 months after the second dose in subjects with a previous natural infection.

TEMRA CD56+ γδ T cell percentages were significantly higher in subjects with a previous natural infection (Figure 4). 

### 3.6. COVID Post-Vaccination and Its Relationship with Immunity

Figure 5 shows the number of subjects who suffered from COVID-19 over the 12 months post-vaccination. Of the 40 subjects recruited in the study, 25 (62.5%) suffered from COVID-19 throughout the study period. Four subjects had COVID-19 twice. No relationship was found between subjects with a previous natural infection prior to vaccination (n = 9) and those with COVID-19 after vaccination (Panels A–D). The evolution of IgG and IgA antibody levels was studied throughout the vaccination period according to COVID-19 post-vaccination (CPV) or not (Figure 5E and Figure 5F, respectively). A significant increase in IgG was observed after the first dose in subjects with CPV. This increase was much greater after the third dose. IgA began to increase after the second dose, reaching significantly higher levels after the third dose in subjects with previous anti-SARS-CoV-2 antibodies. Figure 5G,H shows the number of αβ and γδ T cells prior to vaccination according to CPV or not. Subjects who developed COVID-19 after vaccination had significantly lower baseline γδ T cell levels before vaccination. Subjects who developed COVID-19 after vaccination had significantly lower pre-vaccination levels of αβ and γδ T cell apoptosis (Figure 5I,J). Likewise, the pre-vaccination apoptosis levels of CD19+ cells were also significantly lower in patients who suffered from COVID-19 after vaccination (Figure 5M). No differences in these results were observed when comparing subjects who had a previous natural infection (n = 9) with those who did not have pre-vaccine antibodies from COVID-19 (n = 31). On the other hand, a statistically significant positive correlation was observed between the percentage of γδ T cells and their levels of apoptosis in patients who suffered from COVID-19 after vaccination (Figure 5K,L).

The effectiveness of the vaccines throughout the 12 months of the study was 37.5%. None of the patients who became infected with SARS-CoV-2 after vaccination had severe infection or pneumonia. None of the patients were hospitalized and none died.

### 3.7. Number and Apoptosis at Pre-Vaccine Differentiation Stages of αβ and γδ T-Cell Subsets According to COVID-19 Post-Vaccination

Subjects suffering from CPV had significantly lower pre-vaccine levels of TEM CD56+ and TEMRA CD8+ γδ T cells. However, no differences were observed in the pre-vaccine values of the differentiation stages of αβ T cells. On the other hand, the subjects who suffered from CPV presented significantly lower values in the levels of pre-vaccine apoptosis at all stages of differentiation of αβ T cells. In the case of γδ T cells, pre-vaccine apoptosis was also lower in CPV subjects, with significant differences in the TEMRA CD3+CD8+ subset (Figure 6).

We have not found statistically significant differences in the immune response between the two types of vaccines: Pfizer–BioNTech (BNT162b2) COVID-19 vaccine (mRNA) and Oxford AstraZeneca (ChAdOx1).

## 4. Discussion

In the present work, the immune responses induced by the administration of three doses of the Pfizer–BioNTech (BNT162b2) (mRNA) and Oxford AstraZeneca (ChAdOx1) (adenovirus modified) vaccines were studied. Likewise, the appearance of COVID-19 in the study subjects was evaluated for one year from the first dose of vaccine (Figure 1). It was confirmed, as in other studies, that IgG anti-SARS-CoV-2 antibodies showed a further increase from the second vaccine dose [14,16]. However, subjects without a previous natural infection showed significant increases before the second dose, while subjects with a previous natural infection needed three vaccine doses. This has been verified in very few studies and only with an assessment of IgG anti-SARS-CoV-2-specific antibodies [17]. In our study, subjects without a previous natural infection (specific IgG) showed significant increases after the first dose, while subjects with a previous natural infection only showed significant increases in IgG after the third dose. This could suggest that the vaccine does not stimulate memory cells induced after natural infection; that is, the vaccine only stimulates the response of memory cells induced by its own antigens. Previously, it was shown that early responses to SARS-CoV-2 were dominated by IgA antibodies. IgA remains detectable in saliva for 10 weeks while in the serum, it disappears one month after infection, and IgA dimers, the primary form of antibody in the nasopharynx, were 15 times more potent than IgA monomers against the same antigen. The lack of anti-SARS-CoV-2 IgA and secretory IgA (sIgA) can lead to COVID-19 severity, vaccine failure, and possible prolonged viral shedding [18,19,20]. In our study, the vaccine only produced changes in IgA levels in the subjects without a previous natural infection. They were the ones that presented the highest levels throughout the entire follow-up. The highest IgA responses were observed 8–12 months post-vaccine. This confirms that the highest levels of anti-SARS-CoV-2 antibodies, both IgG and IgA, are achieved after the third vaccination dose. When the dynamics of specific IgA production over time were studied, no significant differences were observed between subjects who suffered from post-vaccination COVID-19 and those who did not. However, an increase in the dynamics of specific IgG production was observed over time, after the first dose, in subjects who suffered from post-vaccination COVID-19. Garcia-Beltran WF et al. observed that the highest levels of IgG and IgA antibodies directed against receptor binding domain and spike were present in critically ill patients who were intubated or died due to COVID-19 [21]. Patients with higher levels of specific IgG could have a higher viral load that increases the risk of symptomatic infection despite vaccination. It has been shown that subjects with a previous SARS-CoV-2 infection generated strong humoral and cellular responses (very strong T cell responses) compared with subjects who had no prior contact with the virus [22,23,24].

It has been shown that CD8 T cells display a protective function from COVID-19 acute disease caused by natural SARS-CoV-2 infection, improving survival. These cytotoxic cells remain active in the tissues for a minimum period of two months [25,26]. Likewise, CD8+ T cells are detectable and functional 7 days after the first vaccine dose when circulating CD4+ T cells and neutralizing antibodies cannot yet be detected [27]. In our study, in addition to the increase in specific IgG in subjects who had anti-SARS-CoV-2 antibodies prior to the vaccine, a significant increase in the number of αβ CD3+CD4+ and CD3+CD8+ γδ T cells was observed six months after the second dose of vaccine. In contrast, apoptosis was decreased in both αβ CD3+CD4+ T cells and CD3+CD8+ and CD3+CD56+ γδ T cells. This demonstrates the important role that γδ T cells play in the vaccinal immunization of previously naturally infected subjects.

However, there was a significant increase in the number and apoptosis of CD3+CD4+ αβ T cells in subjects who did not have previous SARS-CoV-2-specific antibodies. These results demonstrate that subjects naturally exposed to SARS-CoV-2 antigens prior to vaccination behave differently from those who receive the vaccine without prior infection. The decrease in IgG observed six months after the second dose of vaccine corresponds to an increase in the number of CD4+ αβ and CD8+ γδ T cells and a decrease in their apoptosis. This fact could be due to an attempt by the immune system to compensate for this IgG deficit, a fact that does not occur in subjects who did not have previous immunity. This shows that natural immunity acts in a more robust way, probably due to stimulation by all the virus antigens, while the vaccine only induces immunity against the spike protein. This could again indicate that the vaccine does not stimulate memory cells induced after natural infection. In other words, the vaccine would only stimulate the response of memory cells that induce their own antigens.

Knowledge of the differentiation stages of different T-cell subsets is essential to know the situation of activity and reserve that the immune system has in relation to these cells [15]. The study of the different states of activation and memory of T cells has been the subject of this review [28,29,30]. Specific CD4+ T central memory (TCM), CD4+ effector memory (TEM), CD8^+^ TEM, and CD8^+^ terminal effector (TE) cells were all detectable and functional up to 12 months after the second dose of COVID-19 vaccines [31]. In our study, there was a significant increase in naïve CD4+ and CD8+ αβ T cells and naïve CD8+ γδ T cells throughout the vaccination period. In addition, TCM, TEM αβ CD4+, and CD8+ T cells also decreased throughout the study. Similar findings were found in a recent study [32]. However, TEM αβ CD56+ T cells increased at the end of the vaccination process. TEM γδ CD8+ T cells were significantly increased after the second vaccine dose (3rd analysis) in patients who had a previous natural infection (pre-vaccination anti-SARS-CoV-2 antibodies). As will be discussed later, the important role of cytotoxic γδ T cells in maintaining immunity against SARS-CoV-2 is evident. This fact is demonstrated by the great activity and importance of this subset in the mucosa, the entry site of SARS-CoV-2.

Different scenarios of mass COVID-19 vaccination have been studied. While in the best scenario a 95% vaccine efficacy and three years of protection were estimated, in the worse scenario, these data only reached a 50% vaccine efficacy with 45 weeks of protection. These situations can produce dramatical health and economic benefits or damages in the different countries [33,34]. The immunity conferred by primary infection vs. hybrid immunity (immunity developed through a combination of SARS-CoV-2 infection and vaccination) has been studied, demonstrating the greater effectiveness of the latter. In our study, more than 60% of the recruited subjects suffered from COVID-19 infection throughout the 12-month follow-up, although the symptoms were very mild. No patient suffered pneumonia, nor required hospital admission. In addition, no subject died during the study. This confirms that the vaccine does not fully protect against infection, but it does extraordinarily minimize its severity. The effectiveness at 12 months against reinfection in subjects with hybrid immunity is slightly higher than that found in our study in patients with only post-vaccine immunity [35].

In a previous study, the relationship between the decrease in γδ T cells and severity and mortality due to sepsis was demonstrated [36]. Likewise, in hospitalized patients with COVID-19, most severe cases were related with the lowest amount of γδ T cells [37].

In the current study, there was a relationship between the numbers of γδ T cells and the occurrence of SARS-CoV-2 infection after vaccination. The subjects who had COVID-19 post-vaccine had a low frequency of pre-vaccine γδ T-cell subsets compared to those who did not have COVID-19 or did not develop symptoms after infection. To our knowledge, this finding had not been described until now. Interestingly, the stages related to this COVID-19 resistance were TEM and TEMRA cytotoxic γδ T cells. This finding was not observed in the αβ T-cell subsets, which demonstrates the relationship between γδ T cells and the development of immunity against SARS-CoV-2, as well as the immunization mechanisms of the vaccines. In addition, the lymphopenia found in COVID-19 patients has been attributed to SARS-CoV-2-induced activation of apoptosis and the P53 signaling pathway [38]. In the present study, patients who suffered post-vaccination COVID-19 had lower levels of T cell apoptosis. Likewise, a direct correlation between apoptosis and CD3+CD4-CD8- γδ T cells was observed. It therefore seems that in some way, the immune system, given the lower number of γδ T cells in subjects susceptible to subsequent infection by SARS-CoV-2, reduces the apoptosis of these subsets to try to prevent a dramatic decrease in the number of γδ T cells. Further studies are necessary to investigate this assumption.

## 5. Conclusions

Anti-SARS-CoV-2 IgG and IgA antibodies showed a progressive increase during the course of the study. This increase was the greatest after the third dose. The highest levels were observed in subjects who had a previous natural infection (anti-SARS-CoV-2 antibodies) prior to vaccination.

A decrease in specific IgG was observed 6 months after the second dose, at which time there was an increase in CD3+CD4+ αβ, CD3+CD8+ γδ, and TEM CD8+ γδ T cells, and a decrease in apoptosis in CD4+CD8+, CD56+ αβ, and γδ T cells. This fact was manifested in subjects who had a previous natural infection.

Post-vaccination symptomatic COVID-19 infection was greater than 60%. Despite this, the symptoms of the disease were very mild and related to a pre-vaccination deficiency of γδ T cells, specifically CD8+ TEMRA and CD56+ γδ TEM, as well as lower levels of pre-vaccine apoptosis of αβ and γδ T-cell subsets.

The results outline the important role of γδ T cells in SARS-CoV-2-vaccine-mediated protection from the disease.

## Data Availability

The raw data supporting the conclusions of this article will be made available by the authors on request.

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
