# Peer review of "A Low Number of Baselines γδ T Cells Increases the Risk of SARS-CoV-2 Post-Vaccination Infection"

_vaccines, 2024, doi:10.3390/vaccines12050553_

Round 1

Reviewer 1 Report

Comments and Suggestions for Authors

Review of the Article:

Low number of baselines ɣ𝛿 T cells increase the risk of SARS- CoV-2 post-vaccination infection.

This study aimed to investigate the efficacy of Pfizer–BioNTech and Oxford AstraZeneca COVID-19 vaccines over a year, focusing on humoral and cellular immunity and post-vaccination COVID-19 occurrence. Findings revealed a gradual rise in anti-SARS-CoV-2 IgG and IgA antibodies post-vaccination, with the highest increase seen after the third dose, particularly in individuals with pre-existing anti-SARS-CoV-2 antibodies. Additionally, CD4+ αβ, CD8+ γδ, and TEM CD8+ γδ T cell counts increased, while apoptosis in certain T cell subsets decreased. Despite a post-vaccination SARS-CoV-2 infection rate exceeding 60%, COVID-19 symptoms were mild and associated with a deficiency in γδ T cells, specifically CD8+ TEMRA and CD56+ γδ TEM, alongside reduced pre-vaccine apoptosis levels. The authors mention that these results indicate the critical role of γδ T cells in SARS-CoV-2 vaccine-induced immunization.

Minor topics:

Lines 59-60:  modify “…third dose of COVID-19 vaccination with a later a decrease”

With this: “…third dose of COVID-19 vaccination with a later decrease”

Line 133: Is it sample or simple? “2.3. Methods of blood simple analysis”

Line 177: Was used repeated twice :Mann-Whitney U test was used was used to compare”

Major topics:

Box 1: It’s hard to understand, the legend has to explain more, specially the right-hand side of the box.

Did the entire cohort of individuals received all 3 vaccine doses?

What was the number of samples taken in all the time points?

In the text (line 217 -220) you mentioned that you found 5 IgG-positive and 9 IgA-positive individuals but in the box you only mentioned 9? Are these the same individuals?

Figures 2, 3: It is a bit hard to follow, consider separating in A, B C blocks and explain in the legend.

There are very daring statements that are not supported by results depicted. For instance in line 299, the authors indicate that: “The percentage of naïve gd T cell subsets was higher in subjects without previous anti-SARS-CoV-2 antibodies.”

It would also be adequate if the authors pinpoints towards the section of the figure that support those statements.

I strongly recommend to add a section to evaluate if there are differences among the type of vaccine used, Pfizer or AstraZeneca. This could augment the slight differences you are detecting.

In other words, the indistinct use of both vaccines could be introducing variability extra that hampers with the detection of the results, masking interesting signals.

In conclusion:

The article is interesting but it lacks explanation about the individuals. It is hard to follow if they recollect samples from all individuals at all selected time points.

The authors do not discriminate between the use of one vaccine or the other. This deserve an explanation on why they decided to merge together both immunization protocols.

The figures are hard to follow, I recommend divide them into sections A, B, C, etc. Thus, the text should be accompanied by the figures, in other words, if a statement is posted, then a particular section of a figure should be called.

The article uses many of the first-person plural forms like "we did," "we have”. The authors might want to consider using the passive voice more often, such as "it was done" or "it was performed," to give the text a more objective and professional tone.

Comments on the Quality of English Language

The article uses many of the first-person plural forms like "we did," "we have”. The authors might want to consider using the passive voice more often, such as "it was done" or "it was performed," to give the text a more objective and professional tone.

Author Response

Review 1 of the Article:

We appreciate the instructions and we have attended and corrected your suggestions. Thank you very much for your time and comments.

Low number of baselines ɣ? T cells increase the risk of SARS- CoV-2 post-vaccination infection.

This study aimed to investigate the efficacy of Pfizer–BioNTech and Oxford AstraZeneca COVID-19 vaccines over a year, focusing on humoral and cellular immunity and post-vaccination COVID-19 occurrence. Findings revealed a gradual rise in anti-SARS-CoV-2 IgG and IgA antibodies post-vaccination, with the highest increase seen after the third dose, particularly in individuals with pre-existing anti-SARS-CoV-2 antibodies. Additionally, CD4+ αβ, CD8+ γδ, and TEM CD8+ γδ T cell counts increased, while apoptosis in certain T cell subsets decreased. Despite a post-vaccination SARS-CoV-2 infection rate exceeding 60%, COVID-19 symptoms were mild and associated with a deficiency in γδ T cells, specifically CD8+ TEMRA and CD56+ γδ TEM, alongside reduced pre-vaccine apoptosis levels. The authors mention that these results indicate the critical role of γδ T cells in SARS-CoV-2 vaccine-induced immunization.

Minor topics:

Lines 59-60:  modify “…third dose of COVID-19 vaccination with a later a decrease”

With this: “…third dose of COVID-19 vaccination with a later decrease”

Reply: It has been corrected

Line 133: Is it sample or simple? “2.3. Methods of blood simple analysis”

Reply: It is sample. It has been corrected

Line 177: Was used repeated twice: “Mann-Whitney U test was used was used to compare”

Reply: It has been corrected

Major topics:

Box 1: It’s hard to understand, the legend has to explain more, specially the right-hand side of the box.

Did the entire cohort of individuals received all 3 vaccine doses?

What was the number of samples taken in all the time points?

Reply:

All the entire cohort of individuals received 3 vaccine doses. Five blood samples were taken throughout the study as explained in the figure. Second column in yellow.

We have explained it in the Methods.

In the text (line 217 -220) you mentioned that you found 5 IgG-positive and 9 IgA-positive individuals but in the box you only mentioned 9? Are these the same individuals?

Reply:

They are the same individuals. Five subjects were IgG and IgA positive at the same time. It is explained in the text.

“When we analyzed IgG and IgA antibodies against SARS-CoV-2 prior to the first vaccine dose, we found that 5 (12.5%) and 9 (22.5%) patients had IgG and IgA antibodies against SARS-CoV-2, respectively. Five subjects were IgG and IgA positive at the same time”

Figures 2, 3: It is a bit hard to follow, consider separating in A, B C blocks and explain in the legend.

Reply:

Figure 2 has been described as panel A and panel B. Figures 3 and 4 have been described as panels A, B C and D.

There are very daring statements that are not supported by results depicted. For instance in line 299, the authors indicate that: “The percentage of naïve gd T cell subsets was higher in subjects without previous anti-SARS-CoV-2 antibodies.”

It would also be adequate if the authors pinpoints towards the section of the figure that support those statements.

Reply:

The text has been modified and the panel where the assertion is displayed has been indicated

The percentage of naïve gd T cell subsets was higher (not statistically significant) in subjects without previous anti-SARS-CoV-2 antibodies in all differentiation stages (Fig. 4A)”

I strongly recommend to add a section to evaluate if there are differences among the type of vaccine used, Pfizer or AstraZeneca. This could augment the slight differences you are detecting.

In other words, the indistinct use of both vaccines could be introducing variability extra that hampers with the detection of the results, masking interesting signals.

Reply

We have not found significant differences in the immune response between the two types of vaccines: Pfizer–BioNTech (BNT162b2) COVID-19 vaccine (mRNA) and Oxford AstraZeneca (ChAdOx1). We have not described it as it would mean duplicating the figures. But we have added an explanation at the end of the results

“We have not found statistically significant differences in the immune response between the two types of vaccines: Pfizer–BioNTech (BNT162b2) COVID-19 vaccine (mRNA) and Oxford AstraZeneca (ChAdOx1)”.

In conclusion:

The article is interesting but it lacks explanation about the individuals. It is hard to follow if they recollect samples from all individuals at all selected time points.

The authors do not discriminate between the use of one vaccine or the other. This deserve an explanation on why they decided to merge together both immunization protocols.

The figures are hard to follow, I recommend divide them into sections A, B, C, etc. Thus, the text should be accompanied by the figures, in other words, if a statement is posted, then a particular section of a figure should be called.

The article uses many of the first-person plural forms like "we did," "we have”. The authors might want to consider using the passive voice more often, such as "it was done" or "it was performed," to give the text a more objective and professional tone.

Comments on the Quality of English Language

The article uses many of the first-person plural forms like "we did," "we have”. The authors might want to consider using the passive voice more often, such as "it was done" or "it was performed," to give the text a more objective and professional tone.

Reply:

It´s has been corrected

Reviewer 2 Report

Comments and Suggestions for Authors

The main objective of this study was to evaluate in 40 volunteers of the functionality of humoral and cellular immunity after application of the anti-SARS-CoV-2 vaccines (three doses; manufactured by Pfizer-BioNTech and Oxford AstraZeneca) over a period of one year, as well as the occurrence of post-vaccination of COVID-19 disease. In this paper is presented that the rise of IgG and IgA antibodie levels were progressive and it was the greatest after application of the third dose of vaccination. The highest antibodie levels were observed in individuals who had anti-SARS-CoV-2 antibodies prior to vaccination. Also, there was an increase in CD4+ αβ, CD8+ γδ and TEM CD8+ γδ T cells, and a decrease in apoptosis in investigated T cells. The clinical symptoms of post-vaccination COVID-19 disease were mild – they related with deficit of γδ T cells and with lower pre-vaccine apoptosis levels (using 7-AAD) assay). Authors conluded that results obtained in this study show an important role of γδ T cells in SARS-CoV-2 vaccine immunization. mThe study is wide-ranging as well as up to date - I think it undoubtedly deserves to be published in the present form.

Author Response

Review report 2

Dear colleague, we thank you for your favorable review of our work, in which you mention: "The study is comprehensive and up-to-date; I think it certainly deserves to be published in its current format." Thank you for your time and evaluation.

Reviewer 3 Report

Comments and Suggestions for Authors

Review report

The article, written by Andreu-Ballester et al., discusses the importance of γδ T cells in vaccination against SARS-CoV-2. The study analysed humoral and cellular immunity after administration of three doses of the Pfizer-BioNTech and Oxford AstraZeneca vaccines. The results showed a progressive increase in anti-SARS-CoV-2 IgG and IgA antibodies, with higher levels observed in subjects who had anti-SARS-CoV-2 antibodies prior to vaccination. An increase in CD4+ αβ, CD8+ γδ and CD8+ γδ TEM T cells was also observed, as well as decreased apoptosis in CD4+CD8+ and CD56+ αβ and γδ T cells, especially in subjects being infected prior to vacicnation. Post-vaccination SARS-CoV-2 infection was greater than 60 per cent, but COVID-19 symptoms were mild and correlated with γδ T-cell deficiency. 

Major Issues:

-          The abstract needs to be implemented, especially in the ‘Background’ and ‘Methods’ sections. Specifically, the first part needs an in-depth look at current knowledge in the literature, while the second part should include what was examined (IgG, IgA, CD4+/CD8+ T cells) and the methods used.

-          Lines 60-62: I do not agree with these statements, as immune response after booster doses has been well characterized, both against the ‘Wuhan’ strain and currently circulating variants

-          References need to be implemented and updated, as the most recent ones date back to the year 2022. Therefore, they might not consider the latest updates on this specific subject.

-          The same implementation should be applied to the ‘Discussion’ section.

-          I would replace the 'we + action' form with the passive form throughout all the text, especially in M&M section.  E.g. lines 119-121 should be converted as ‘An enrichment of the sample in mononuclear cells (MNCs) was obtained by centrifuging the EDTA anticoagulated blood sample on a density gradient using Lymphoprep™ (Palex Medical SA) at 3,500 rpm for 20 minutes’.

-          There are many graphics in the text, certainly a sign of the extensive work done. However, so as not to confuse the reader, as they are perhaps excessive in number, I would consider it appropriate to include only those where significant differences are present. The others could be included in a Supplementary Appendix.

-          For the figures that will remain in the main article, I would suggest adding a numbering for each one to prevent the reader from getting lost (e.g. Figure 2).

Minor issues:

-          In the text, reference is made to subjects having antibodies prior to vaccination, rather I would refer to this category by labelling them as 'subjects with previous natural infection' or ‘with previous natural infection’.

-          Line 35: I would suggest removing ‘wrongly known as the Spanish Flu’.

-          Line 44: I would suggest adding more references.

-          Lines 48-57: text is poor in references; I would add some more.

-          Line 58: neutralizing antibodies.

-          Line 60: ‘issues’ would be preferable instead of ‘unknowns’.

-          Lines 103-104: I do not understand this statement. Does it mean that, for example, the third withdrawal (6 months after the second dose) took place the day before the third dose was administered? I apologise for the misunderstanding.

-          Paragraph 2.3: I would avoid dividing the methods below into sub-chapters, thus converting chapter 2.3.1 into 2.3, 2.3.2 into 2.4 and so on.

-          Paragraph 2.3.2: ‘clone’ instead of ‘clon’. Use passive form (lines 139-144).

-          I would remove the 'highlights' section in the results because it is unnecessary.

-          Paragraph 3.1: Were differences in the immune response been observed according to gender and age of the subjects? If yes, it would be appreciated to include them.

-          Lines 220-221: ‘during the course of the study’ instead of ‘vaccination period’.

-          Lines 226-227: see the beginning of ‘minor issues’ section.

-          Line 253: I would write 'returned to baseline levels' or 'decreased to baseline levels' after the third dose rather than 'decreased from the third dose'.

-          Line 331-332: ‘before vaccination’ means ‘respect to those who had COVID-19 before vaccination’?

-          Figure 5: please revise the caption as it is not clear to follow (especially A-D). Panel F, J, M caption missing.

Author Response

Review report 3

Thank you for your suggestions and your time.

The article, written by Andreu-Ballester et al., discusses the importance of γδ T cells in vaccination against SARS-CoV-2. The study analysed humoral and cellular immunity after administration of three doses of the Pfizer-BioNTech and Oxford AstraZeneca vaccines. The results showed a progressive increase in anti-SARS-CoV-2 IgG and IgA antibodies, with higher levels observed in subjects who had anti-SARS-CoV-2 antibodies prior to vaccination. An increase in CD4+ αβ, CD8+ γδ and CD8+ γδ TEM T cells was also observed, as well as decreased apoptosis in CD4+CD8+ and CD56+ αβ and γδ T cells, especially in subjects being infected prior to vacicnation. Post-vaccination SARS-CoV-2 infection was greater than 60 per cent, but COVID-19 symptoms were mild and correlated with γδ T-cell deficiency. 

Major Issues:

-          The abstract needs to be implemented, especially in the ‘Background’ and ‘Methods’ sections. Specifically, the first part needs an in-depth look at current knowledge in the literature, while the second part should include what was examined (IgG, IgA, CD4+/CD8+ T cells) and the methods used.

Reply: Background and Methods have been modified

Background: The COVID-19 pandemic is the biggest global health problem in the last hundred years. The effectiveness of the vaccine is estimated between 70-95% according to the studies carried out, although there are aspects of the immune response to the vaccine that remain unclear.

Methods: The humoral and cellular immunity after the administration of three doses of the Pfizer–BioNTech and Oxford AstraZeneca vaccines against SARS-CoV-2 over one year and the appearance of post-vaccination COVID-19 have been studied. SARS-CoV-2 IgG and IgA antibodies, αβ and γδ T cell subsets and their differentiation stages and apoptosis were analyzed.

-          Lines 60-62: I do not agree with these statements, as immune response after booster doses has been well characterized, both against the ‘Wuhan’ strain and currently circulating variants 

The text has been corrected

“Although a significant decrease in the humoral and cellular response after 6 months of vaccination was recently published [7]. There are still many questions to be resolved about COVID-19 vaccines, such as the duration of humoral immunity after primary infection or long-term vaccination. Similarly, knowledge about the duration of protection conferred by booster injections is also limited [8]”.

-         References need to be implemented and updated, as the most recent ones date back to the year 2022. Therefore, they might not consider the latest updates on this specific subject. 

Reply:

Reference added

[7] Favresse J, Gillot C, Closset M, Cabo J, Wauthier L, David C, Elsen M, Dogné JM, Douxfils J. Durability of humoral and cellular immunity six months after the BNT162b2 bivalent booster. J Med Virol. 2024 Jan;96(1):e29365. doi: 10.1002/jmv.29365.

-          The same implementation should be applied to the ‘Discussion’ section. 

Reply:

References added

  1. Wang Z, Lorenzi JCC, Muecksch F, Finkin S, Viant C, Gaebler C, Cipolla M, Hoffmann HH, Oliveira TY, Oren DA, Ramos V, Nogueira L, Michailidis E, Robbiani DF, Gazumyan A, Rice CM, Hatziioannou T, Bieniasz PD, Caskey M, Nussenzweig MC. Enhanced SARS-CoV-2 neutralization by dimeric IgA. Sci Transl Med. 2021 Jan 20;13(577):eabf1555. doi: 10.1126/scitranslmed.abf1555.

  1. Quinti I, Mortari EP, Fernandez Salinas A, Milito C, Carsetti R. IgA Antibodies and IgA Deficiency in SARS-CoV-2 Infection. Front Cell Infect Microbiol. 2021 Apr 6;11:655896. doi: 10.3389/fcimb.2021.655896.

-          I would replace the 'we + action' form with the passive form throughout all the text, especially in M&M section.  E.g. lines 119-121 should be converted as ‘An enrichment of the sample in mononuclear cells (MNCs) was obtained by centrifuging the EDTA anticoagulated blood sample on a density gradient using Lymphoprep™ (Palex Medical SA) at 3,500 rpm for 20 minutes’.

Reply:

It has been corrected

-          There are many graphics in the text, certainly a sign of the extensive work done. However, so as not to confuse the reader, as they are perhaps excessive in number, I would consider it appropriate to include only those where significant differences are present. The others could be included in a Supplementary Appendix.

Reply:

We understand the reviewer's suggestion, but there are statistically significant elements in all the graphs. Eliminating from the graph those panels that are not significant, we understand that it would not give a global and continuous vision of the specific study.

-          For the figures that will remain in the main article, I would suggest adding a numbering for each one to prevent the reader from getting lost (e.g. Figure 2).

Reply:

It has been corrected

Minor issues: 

-          In the text, reference is made to subjects having antibodies prior to vaccination, rather I would refer to this category by labelling them as 'subjects with previous natural infection' or ‘with previous natural infection’. 

Reply:

It has been corrected in all text

-          Line 35: I would suggest removing ‘wrongly known as the Spanish Flu’.

Reply:

“known as the Spanish flu” has been eliminated

-          Line 44: I would suggest adding more references.

Reply:

References added

  1. Gutfreund MC, Kobayashi T, Callado GY, Pardo I, Hsieh MK, Lin V, Perencevich EN, Salinas JL, Edmond MB, Mendonça E, Rizzo LV, Marra AR. The effectiveness of the COVID-19 vaccines in the prevention of post-COVID conditions in children and adolescents: a systematic literature review and meta-analysis. Antimicrob Steward Healthc Epidemiol. 2024 Apr 19;4(1):e54. doi: 10.1017/ash.2024.42.

  1. Zhu C, Pang S, Liu J, Duan Q. Current Progress, Challenges and Prospects in the Development of COVID-19 Vaccines. Drugs. 2024 Apr 23. doi: 10.1007/s40265-024-02013-8.

-          Lines 48-57: text is poor in references; I would add some more.

Reply:

References added

  1. Campo F, Venuti A, Pimpinelli F, Abril E, Blandino G, Conti L, De Virgilio A, De Marco F, Di Noia V, Di Domenico EG, Di Martino S, Ensoli F, Giannarelli D, Mandoj C, Mazzola F, Moretto S, Petruzzi G, Petrone F, Pichi B, Pontone M, Vidiri A, Vujovic B, Piaggio G, Sperandio E, Rosati V, Cognetti F, Morrone A, Ciliberto G, Pellini R. Antibody Persistence 6 Months Post-Vaccination with BNT162b2 among Health Care Workers. Vaccines (Basel). 2021 Oct 3;9(10):1125. doi: 10.3390/vaccines9101125.
  2. Klompas M. Understanding Breakthrough Infections Following mRNA SARS-CoV-2 Vaccination. JAMA. 2021 Nov 23;326(20):2018-2020. doi: 10.1001/jama.2021.19063.

-          Line 58: neutralizing antibodies.

Reply:

It has been corrected

-          Line 60: ‘issues’ would be preferable instead of ‘unknowns’. 

Reply:

The word has been corrected

-          Lines 103-104: I do not understand this statement. Does it mean that, for example, the third withdrawal (6 months after the second dose) took place the day before the third dose was administered? I apologise for the misunderstanding.

Reply:

Yes, each analysis was performed immediately before the vaccine administration, on the same day.

-          Paragraph 2.3: I would avoid dividing the methods below into sub-chapters, thus converting chapter 2.3.1 into 2.3, 2.3.2 into 2.4 and so on.

Reply:

The chapters numeration has been transformed

-          Paragraph 2.3.2: ‘clone’ instead of ‘clon’. Use passive form (lines 139-144).

Reply:

Clon has been replaced by clone

-          I would remove the 'highlights' section in the results because it is unnecessary.

Reply:

Highlights has been removed

-          Paragraph 3.1: Were differences in the immune response been observed according to gender and age of the subjects? If yes, it would be appreciated to include them. 

Reply:

To carry out the study by age groups, there is not enough sample of subjects. Regarding sex, we have not found differences

-          Lines 220-221: ‘during the course of the study’ instead of ‘vaccination period’.

Reply:

It has been replaced

-          Lines 226-227: see the beginning of ‘minor issues’ section.

Reply:

It has been corrected

-          Line 253: I would write 'returned to baseline levels' or 'decreased to baseline levels' after the third dose rather than 'decreased from the third dose'.

Reply:

It has been corrected

-          Line 331-332: ‘before vaccination’ means ‘respect to those who had COVID-19 before vaccination’?

Reply:

The analysis includes subjects with previous natural infection or not. We analyzed both groups and no significant differences were found

-          Figure 5: please revise the caption as it is not clear to follow (especially A-D). Panel F, J, M caption missing.

Reply:

The legend of figure 5 has been corrected.

Figure 5. A: Relation between COVID-19 at some point in the vaccination period (COVID-19 Post-vaccine –CPV-) (n=25) and pre-vaccination anti-SARS-CoV-2. B: Post-vaccine COVID-19 after 1st dose. C: Post-vaccine COVID-19 after 2nd dose. D: Post-vaccine COVID-19 after 3nd dose. E and F: IgG and IgA according to CPV (COVID-19 Post-vaccine) respectively. G and H: Pre-vaccine αβ-gd T cell subsets number according to CPV (COVID-19 Post-vaccine). I and J: αβ-gd T cells subsets APOPTOSIS pre-vaccine according to CPV (COVID-19 Post-vaccine). U test was used to compare differences between CPV and No. Significance: *p<0.05, **p<0.01. K and L: Relation between gd T cells and apoptosis (n=25) (Pearson Test was used). DN=Double Negative (CD4-CD8-). M: Pre-vaccine B (CD19+) cells subsets number according to CPV (COVID-19 Post-vaccine).

Round 2

Reviewer 3 Report

Comments and Suggestions for Authors

All suggested corrections were addressed correctly and comprehensively.